# A rapid, specific, extraction-less, and cost-effective RT-LAMP test for the detection of SARS-CoV-2 in clinical specimens

**Francesco Elia Marino** *, Eric Proffitt, Eugene Joseph, Arun Manoharan

Department of Research and Development, Prime Discoveries INC, Philadelphia, Pennsylvania, United States of America

* francescoelia.marino@gmail.com

**Data Availability Statement:** All relevant data are within the manuscript and its Supporting information files.

## Abstract

In 2019 a newly identified coronavirus, designated as severe acute respiratory syndrome coronavirus 2 (SARS-CoV-2), has spread rapidly from the epicenter in Wuhan (China) to more than 150 countries around the world, causing the Coronavirus disease 2019 (COVID-19) pandemic. In this study, we describe an extraction-less method based on reverse transcriptase loop-mediated isothermal amplification (RT-LAMP) intended for the rapid qualitative detection of nucleic acid from SARS-CoV-2 in upper respiratory specimens, including oropharyngeal and nasopharyngeal swabs, anterior nasal and mid-turbinate nasal swabs, nasopharyngeal washes/aspirates or nasal aspirates as well as bronchoalveolar lavage (BAL) from individuals suspected of COVID-19 by their healthcare provider. The assay's performance was evaluated and compared to an RT quantitative PCR-based assay (FDA-approved). With high sensitivity, specificity, and bypassing the need for RNA extraction, the RT-LAMP Rapid Detection assay is a valuable and fast test for an accurate and rapid RNA detection of the SARS-CoV-2 virus and potentially other pathogens. Additionally, the versatility of this test allows its application in virtually every laboratory setting and remote location where access to expensive laboratory equipment is a limiting factor for testing during pandemic crises.

## Introduction

In December 2019, an outbreak in Wuhan of a severe respiratory illness was caused by a previously unrecognized coronavirus, which has since been named severe acute respiratory syndrome coronavirus 2 (SARS-CoV-2) [1–6]. After the virus spread in more than 150 countries worldwide, the COVID-19 was declared a worldwide pandemic. The SARS-CoV-2 related pandemic has posed challenges for the global market, economy, and scientific research and underlined significant inequality and inaccessibility to testing for many countries worldwide [7,8]. Point-of-care serial screening can provide rapid results, and it is critical to identify asymptomatic individuals carrying the virus. Classic methods of screening, and virus RNA detection like RT-PCR, are labor expensive, require additional reagents for RNA extraction,

**Funding:** The authors received no specific funding for this work. The authors received salaries from Prime Discoveries Inc. Prime Discoveries is a venture funded company. The venture capitalist funders provided salaries for authors [FEM, EP, EJ, AM]. however, they did not have any additional role in the study design, data collection, analysis, publication decision, or manuscript preparation. The specific roles of these authors are articulated in the 'author contributions' section.

**Competing interests:** The Prime Covid DetectTM Rapid Detection Assay is offered as a catalog item at Prime Discoveries Inc. The pipeline used for the SARS-CoV-2 detection in clinical samples using an RNA-extraction less protocol and RT-LAMP (Prime Covid DetectTM Rapid Detection Assay) is part of the patent application number 63/185,571 "An isothermal amplification system and method of use of the same for detecting pathogenic infection in a subject" (patent status: pending). The Prime Covid DetectTM Rapid Detection kit (cat# PRDICOV Prime Discoveries) is registered as In Vitro Medical Device in the European Union - CE approval (Certificate number: 2022-IVD/CE356). None of the points above alter the authors' adherence to PLOS One policies on sharing data and materials.

and highly trained technicians in molecular biology techniques are needed. RT-LAMP (loop-mediated isothermal amplification) methods based on a colorimetric read-out have been evaluated as a suitable alternative to the regular PCR methods [9]. However, samples require refrigeration and must be analyzed within a short time frame. Additionally, RT-LAMP combined with a colorimetric read-out poses challenges for data interpretation due to high ambiguity and pH fluctuations can easily alter the readout [10].

Additionally, the current pandemic has introduced an unprecedented challenging situation in obtaining plastic consumables and RNA extraction kits. Manufacturers worldwide still have issues satisfying the demand for highly requested reagents for SARS-CoV-2 serial screening and biomedical research in general. The crisis has also accentuated significant disparities, and laboratories in remote locations and with limited budgets can hardly afford expensive quantitative PCR equipment and reagents. Rapid tests for SARS-CoV-2 were implemented as fast assays to control transmission and provide results in less than one hour. However, increasing evidence suggests significant risks associated with the false positivity and negativity rate of these tests, especially if the screening strategy is based on lateral flow antigen tests (rapid tests) [11,12]. Thus, alternative testing methods based on fast and reliable approaches without compromising the rigor of the testing pipeline are critically needed.

The current study shows an extraction-less method that can go from patient sample collection to testing and data interpretation in less than one hour (Prime CovidDetect™ Rapid Detection kit). This method is based on isothermal amplification and can be performed virtually on every equipment able to maintain a temperature of 65 degrees Celsius for 50 minutes and can be paired with any plate reader and does not necessarily need quantitative PCR equipment for data interpretation and visualization. The assay utilizes the well-established LAMP (loop-mediated isothermal amplification) method [6,13–15] combined with the innovative iSWAB™ Extraction-less buffer (Mawi DNA Technologies) that was designed to eliminate the RNA extraction step in the COVID-19 Molecular testing workflow, allowing researchers to perform direct RT-PCR or RT-LAMP on individual and pooled samples. The iSWAB- Extraction-less buffer is a non-toxic stabilizing technology that enables the inactivation of bacteria, fungi, spores, and viruses, allows ambient collection and transport of various bio-samples, and preserve the nucleic acid material at the time of collection. The buffer stabilizes nasal swabs, and these samples can be used directly in the RT-LAMP reactions without any prior major (RNA extraction) or minor (heating or/and Proteinase K treatment) sample processing and successfully detect SARS-CoV-2 without any observed PCR inhibition. After assessing the Limit of Detection (LoD) and comparing the detection rate, sensitivity, specificity, of the Prime CovidDetect™ Rapid Detection kit to a standard comparator assay (FDA approved) for the detection of SARS-CoV-2, we showed that the assay is a robust alternative to PCR based assays and can be virtually adopted in any laboratory settings for the rapid identification of the SARS-CoV-2 RNA.

## Material and methods

### LAMP reagents and reaction set-up

The Prime CovidDetect™ Rapid Detection kit used eighteen primers to identify SARS-CoV-2 (ORF1 a/b, E, and N genes) and six primers to identify the 18S Ribosomal RNA gene (18S RNA) used as control. The primers' sequences are illustrated in Table 1. Primers were mixed to obtain the following final concentrations per reaction (FIP 0.8 μM, BIP 0.8 μM, F3 0.1 μM, B3 0.1 μM, LB 0.2 μM, LF 0.2 μM). LAMP enzyme and Dye were purchased from New England Biolabs (cat# E1700L). The reaction set-up was prepared as follows: 4 μl of Enzyme Mix, 0.5 μl of SARS-CoV-2 Primers Mix (from 20X Stock) or 0.5 μl of 18S RNA Primers Mix

**Table 1. LAMP primers.**

| Primer Name | Sequence (5'-3') |
| --- | --- |
| **N Gene** | |
| N2-F3 | ACCAGGAACTAATCAGACAAG |
| N2-B3 | GACTTGATCTTTGAAATTTGGATCT |
| N2-FIP | TTCCGAAGAACGCTGAAGCGGAACTGATTACAAACATTGGCC |
| N2-BIP | CGCATTGGCATGGAAGTCACAATTTGATGGCACCTGTGTA |
| N2-LF | GGGGGCAAATTGTGCAATTTG |
| N2-LB | CTTCGGGAACGTGGTTGACC |
| **E Gene** | |
| E1-F3 | TGAGTACGAACTTATGTACTCAT |
| E1-B3 | TTCAGATTTTTAACACGAGAGT |
| E1-FIP | ACCACGAAAGCAAGAAAAAGAAGTTCGTTTCGGAAGAGACAG |
| E1-BIP | TTGCTAGTTACACTAGCCATCCTTAGGTTTTACAAGACTCACGT |
| E1-LF | CGCTATTAACTATTAACG |
| E1-LB | GCGCTTCGATTGTGTGCGT |
| **ORF1 gene** | |
| ORF1-F3 | CGGTGGACAAATTGTCAC |
| ORF1-B3 | CTTCTCTGGATTTAACACACTT |
| ORF1-FIP | TCAGCACACAAAGCCAAAAATTTATTTTTCTGTGCAAAGGAAATTAAGGAG |
| ORF1-BIP | TATTGGTGGAGCTAAACTTAAAGCCTTTTCTGTACAATCCCTTTGAGTG |
| ORF1-LF | TTACAAGCTTAAAGAATGTCTGAACACT |
| ORF1-LB | TTGAATTTAGGTGAAACATTTGTCACG |
| **18S RNA** | |
| 18S RNA-F3 | GTTCAAAGCAGGCCCGAG |
| 18S RNA-B3 | CCTCCGACTTTCGTTCTTGA |
| 18S RNA-FIP | TGGCCTCAGTTCCGAAAACCAACCTGGATACCGCAGCTAGG |
| 18S RNA-BIP | GGCATTCGTATTGCGCCGCTGGCAAATGCTTTCGCTCTG |
| 18S RNA-LF | AGAACCGCGGTCCTATTCCATTATT |
| 18S RNA-LB | ATTCCTTGGACCGGCGCAAG |

(from 20X Stock), 0.25 μl of LAMP Dye, 3 μl of Input RNA (from the iSWAB™ Extraction-less buffer), 2.25 of Nuclease-Free Water (final reaction volume was 10 μl).

For each sample, four reactions were prepared: two replicates were prepared to detect SARS-CoV-2 and two for the detection of 18S RNA. Samples were loaded into a 384-well plate (cat#4309849 Thermo Fisher Scientific), sealed with optical adhesive film (cat# 4311971 Fisher Scientific), and centrifuged at 1000 rpm for 1 minute. The QuantStudio™ 5 was used to set up the reaction as follows: 100 cycles (each cycle of 30 seconds incubated at 65 degrees Celsius) were selected as PCR steps (on the FAM channel), and data collection was set to ON (for data

**Table 2. Cut-off values.**

| Control Type | RT LAMP | |
| --- | --- | --- |
| | **ORF1, E, N (FAM channel)** | **18S RNA (FAM channel)** |
| Negative | Non-detected or detection $\geq 80$ cycles | Non-detected or detection $\geq 80$ cycles |
| Positive | Detection $\leq 80$ cycles | Detection $\leq 80$ cycles |

**Table 3. In silico inclusivity analysis.**

|  | N-gene | E-gene | ORF1 region |
|---|---|---|---|
| **Total Primer Length (nt)** | 169 | 168 | 187 |
| **Total # of Strains Evaluated** | 5773195 | 5773195 | 5773195 |
| **100% Match** | 5341972 | 4612424 | 5473635 |
| **1 Mismatch** | 409281 | 1147905 | 276374 |
| **2 Mismatches** | 17108 | 7490 | 17235 |
| **3 Mismatches** | 919 | 128 | 637 |
| **>3 Mismatches** | 3915 | 5248 | 5314 |

collection). Cut-off values were applied as illustrated in Table 2. See the S1 File for a step-by-step instrument set-up and alternative instruments that could be used with the assay.

## Analytical sensitivity

Quantified heat-inactivated SARS-CoV-2 virus (cat#VR-1986 ATCC Lot# 70042082–3.9 x $10^5$ genome copies/ml) was spiked into a real clinical matrix (nasopharyngeal swabs from 10 negative samples collected in iSWAB™ Extraction-less buffer) and used for serial dilutions. The LoD concentration was determined by testing 24 individual replicates for different dilutions (as recommended by the FDA). LoD was defined as the lowest concentration at which more than 95% of replicates were positive. Replicates were called negative if no amplification was detected before cycle 80 (threshold value established based on nonspecific amplification observed for detection at cycles $\geq$ 80) of the RT-LAMP according to the assay selecting criteria to call a sample positive or negative. Homology analysis was conducted for the ORF1, E, and N, primer sets against all SARS-CoV-2 sequences deposited at GISAID [16–18] on March 16, 2022. A total of 9,308,692 sequences were considered, of which 3,535,497 were discarded for being incomplete ($\leq$ 29kb) or having poor coverage ($\geq$ 1% undefined bases). The remaining 5,773,195 sequences comprise a superset of those sequences considered by GISAID to be both complete and high-coverage (GISAID evaluates genomes >29,000bp as complete and further assigns labels of high coverage <1% Ns—undefined bases- and low coverage >5% Ns. The exact locations of the primer regions in each sequence were identified from the multiple sequence alignment file provided by GISAID. Subsequently, for each of the three primer sets, the number of mismatches per sequence was calculated using the Levenshtein distance metric [19] Table 3.

## Analytical specificity

In silico cross-reactivity analysis was performed by aligning the SARS-CoV-2 primer sequences against sequences of common viruses as well as those coronaviruses most closely related to SARS-CoV-2. See Table 4 for the organisms assessed in silico for potential cross-reactivity. The analytical specificity was also assessed by wet testing. Briefly, samples were prepared by spiking intact viral particles or cultured RNA or bacterial cells into real clinical matrix as described before using panels/organisms from Zeptometrix, BEI Resources, and ATCC Table 5. Because no quantification information was available for the individual wet tested organisms, 50 μL of each stock was spiked into a negative clinical matrix and tested in replicates of three.

**Table 4.** *In Silico* cross-reactivity/exclusivity.

| GenBank | Designation | N-gene | E-gene | Orf1 region |
|---|---|---|---|---|
| MN908947.3 | Severe acute respiratory syndrome coronavirus 2 isolate Wuhan-Hu-1, complete genome | 100.00% | 100.00% | 100.00% |
| NC_002645.1 | Human coronavirus 229E, complete genome | 56.80% | 54.80% | 47.60% |
| NC_006213.1 | Human coronavirus OC43 strain ATCC VR-759, complete genome | 49.70% | 48.80% | 44.40% |
| NC_006577.2 | Human coronavirus HKU1, complete genome | 46.20% | 50.00% | 48.70% |
| NC_005831.2 | Human Coronavirus NL63, complete genome | 57.40% | 55.40% | 48.10% |
| NC_004718.3 | SARS coronavirus Tor2, complete genome | 82.20% | 96.40% | 44.40% |
| NC_019843.3 | Middle East respiratory syndrome-related coronavirus isolate HCoV-EMC/2012, complete genome | 48.50% | 54.20% | 47.60% |
| X67709.1 | Adenovirus type 1 hexon gene | 13.60% | 13.70% | 31.00% |
| NC_039199.1 | Human metapneumovirus isolate 00–1, complete genome | 43.80% | 54.80% | 47.10% |
| AF457102.1 | HPIV-1 strain Washington/1964, complete genome | 53.80% | 50.60% | 48.10% |
| AF533012.1 | Human parainfluenza virus 2 strain GREER, complete genome | 45.00% | 50.00% | 49.20% |
| KF530234.1 | Human parainfluenza virus 3 strain HPIV3/MEX/1526/2005, complete genome | 48.50% | 50.60% | 46.50% |
| NC_021928.1 | Human parainfluenza virus 4a viral cRNA, complete genome, strain: M-25 | 46.70% | 54.80% | 45.50% |
| FJ966079.1 | Influenza A virus (A/California/04/2009(H1N1)) segment 1 polymerase PB2 (PB2) gene, complete cds | 44.40% | 34.50% | 34.20% |
| KT002533.1 | Influenza A virus (A/canine/Illinois/12191/2015(H3N2)) segment 1 polymerase PB2 (PB2) gene, complete cds | 37.30% | 32.70% | 23.50% |
| MN230203.1 | Influenza B virus (B/California/24/2019) segment 1 polymerase PB1 (PB1) gene, complete cds | 29.00% | 23.80% | 29.40% |
| MK715533.1 | Influenza B virus (B/California/40/2018) segment 1 polymerase PB1 (PB1) gene, complete cds | 35.50% | 41.70% | 37.40% |
| KP745766.1 | Enterovirus D68 isolate NY328, complete genome | 45.00% | 41.70% | 42.80% |
| U39661.1 | Respiratory syncytial virus, complete genome | 49.70% | 50.00% | 50.30% |
| NC_001490.1 | Rhinovirus B14, complete sequence | 45.60% | 44.60% | 44.40% |

## Clinical samples

Positive (n = 30) and negative (n = 34) nasopharyngeal swabs were purchased from LEE BioSolutions and placed in the MAWI iSWAB™ Extraction-less buffer. The manufacturer confirmed samples' negative or positive status using the TaqPath COVID-19 combo kit (cat# A47814 Thermo Fisher Scientific). The samples' status (negative or positive) was re-confirmed by using the FDA-approved Quick-SARS-CoV-2 rRT-PCR kit (cat# R3011 Zymo Research) following the manufacturer's instructions. According to the manufacturer's provided information, the symptomatic status of the patients was unknown at the time of collection. Thus, an additional set of samples with known patients' symptomatic status, positive symptomatic (n = 32), positive asymptomatic (n = 36), and negative (n = 49) were obtained from a diagnostic lab (Hook Diagnostics) and re-confirmed using the Quick-SARS-CoV-2 rRT-PCR kit (cat# R3011 Zymo Research). These samples were collected from 7 testing sites across the United States.

## RNA extraction

For samples to be analyzed with the comparator assay (Quick-SARS-CoV-2 rRT-PCR kit), 140 μL of input material (nasopharyngeal swab in iSWAB™ extraction-less MAWI buffer) was used for RNA extraction performed with the QIAamp Viral RNA kit (cat# 52906 Qiagen) according to the manufacturer's instructions except for the final elution step (performed in 20 μL of AVE buffer instead of 60 μL).

## Results

### Sensitivity and specificity of the assay

To assess the sensitivity of the assay we firstly investigated the limit of detection (LoD) to define the lowest limit at which the assay can detect the presence of intact virus with

**Table 5. Cross-reactivity/exclusivity wet testing of the Prime CovidDetect™ rapid detection kit.**

| Organism | Strain | Provider | Catalog number | ORF1/N/E-gene Detected Replicates |
|---|---|---|---|---|
| Adenovirus 11 | Slobitski | ATCC | VR-12 | 0/3 |
| Adenovirus 5 | Adenoid 75 | ATCC | VR-5 | 0/3 |
| *Bordetella pertussis* | 18323 [NCTC 10739] | ATCC | 9797 | 0/3 |
| *Candida albicans* | NIH 3172 | ATCC | 14053 | 0/3 |
| *Chlamydophila pneumoniae* | TWAR strain 2023 | ATCC | VR-1356 | 0/3 |
| Enterovirus 70 | J670/71 | ATCC | VR-836 | 0/3 |
| *Haemophilus influenzae* | NCTC 8143 | ATCC | 33391 | 0/3 |
| Human parainfluenza virus 4b | CH 19503 | ATCC | VR-1377 | 0/3 |
| Human respiratory syncytial virus | A2 | ATCC | VR-1540P | 0/3 |
| Human rhinovirus 61 | 6669-CV39 [V-152-002-021] | ATCC | VR-1171 | 0/3 |
| *Mycobacterium tuberculosis* | H37Ra | ATCC | 25177 | 0/3 |
| *Mycoplasma pneumoniae* | Somerson et al. FH strain of Eaton Agent [NCTC 10119] | ATCC | 15531 | 0/3 |
| *Pseudomonas aeruginosa* | (Schroeter) Migula (ATCC® 10145™)—[CCEB 481, MDB strain BU 277, NCIB 8295, NCPPB 1965, NCTC 10332, NRRL B-771, R. Hugh 815] | ATCC | 10145 | 0/3 |
| *Staphylococcus epidermidis* | AmMS 205 | ATCC | 49134 | 0/3 |
| *Streptococcus pneumoniae* | Mu50 [NRS1] | ATCC | 700699 | 0/3 |
| *Streptococcus pyogenes* | Rosenbach (ATCC® 49399™–QC A62) | ATCC | 49399 | 0/3 |
| *Streptococcus salivarius* | B2 | ATCC | 9759 | 0/3 |
| Human coronavirus | | BEI | NL63 | 0/3 |
| Human coronavirus | | BEI | 229E | 0/3 |
| Human coronavirus, Middle East Respiratory Syndrome Coronavirus (MERS-CoV), | EMC/2012 | BEI | NR-50549 | 0/3 |
| SARS Coronavirus | | BEI | NR-3882 | 0/3 |
| SARS-Related Coronavirus 2 | | BEI | NR-52286 | 0/3 |
| A. baumannii | 307–0294 | ZeptoMetrix | NATPPQ-BIO | 0/3 |
| Adenovirus Type 3 | | ZeptoMetrix | NATRVP-1 | 0/3 |
| Adenovirus Type 3 | | ZeptoMetrix | NATPPA-BIO | 0/3 |
| Adenovirus Type 31 | | ZeptoMetrix | NATPPA-BIO | 0/3 |
| C. pneumoniae | CWL-029 | ZeptoMetrix | NATPPA-BIO | 0/3 |
| Coronavirus | 229E | ZeptoMetrix | NATRVP-1 | 0/3 |
| Coronavirus | NL63 | ZeptoMetrix | NATPPA-BIO | 0/3 |
| Coronavirus | OC43 | ZeptoMetrix | NATRVP-1 | 0/3 |
| Coronavirus | SARS | ZeptoMetrix | NATRVP-1 | 0/3 |
| E. cloacae | Z101 | ZeptoMetrix | NATPPQ-BIO | 0/3 |
| E. coli | Z297 | ZeptoMetrix | NATPPQ-BIO | 0/3 |
| Enterovirus | | ZeptoMetrix | NATRVP-1 | 0/3 |
| H. influenzae | MinnA | ZeptoMetrix | NATPPQ-BIO | 0/3 |
| Human Metapneumovirus | | ZeptoMetrix | NATRVP-1 | 0/3 |
| Influenza A | H1 | ZeptoMetrix | NATRVP-1 | 0/3 |
| Influenza A | H1N1 (2009) | ZeptoMetrix | NATRVP-1 | 0/3 |
| Influenza A | H3 | ZeptoMetrix | NATRVP-1 | 0/3 |
| Influenza A | H3 A/Brisbane/10/07 | ZeptoMetrix | NATPPA-BIO | 0/3 |
| Influenza B | | ZeptoMetrix | NATRVP-1 | 0/3 |
| Influenza B | B/Florida/02/06 | ZeptoMetrix | NATPPA-BIO | 0/3 |
| K. aerogenes | Z052 | ZeptoMetrix | NATPPQ-BIO | 0/3 |

*(Continued)*

**Table 5.** (Continued)

| Organism | Strain | Provider | Catalog number | ORF1/N/E-gene Detected Replicates |
|---|---|---|---|---|
| K. oxytoca | Z115 | ZeptoMetrix | NATPPQ-BIO | 0/3 |
| K. pneumoniae | KPC2 | ZeptoMetrix | NATPPQ-BIO | 0/3 |
| K. pneumoniae | Z138; OXA-48 | ZeptoMetrix | NATPPQ-BIO | 0/3 |
| K. pneumoniae | Z460; NDM-1 | ZeptoMetrix | NATPPQ-BIO | 0/3 |
| L. pneumophila | Philadelphia | ZeptoMetrix | NATPPA-BIO | 0/3 |
| M. catarrhalis | Ne 11 | ZeptoMetrix | NATPPQ-BIO | 0/3 |
| M. pneumoniae | M129 | ZeptoMetrix | NATPPA-BIO | 0/3 |
| Metapneumovirus | 8 Peru6-2003 | ZeptoMetrix | NATPPA-BIO | 0/3 |
| P. aeruginosa | Z139, VIM-1 | ZeptoMetrix | NATPPQ-BIO | 0/3 |
| P. mirabilis | Z050 | ZeptoMetrix | NATPPQ-BIO | 0/3 |
| Parainfluenza virus Type 1 | | ZeptoMetrix | NATPPA-BIO | 0/3 |
| Parainfluenza virus Type 1 | | ZeptoMetrix | NATRVP-1 | 0/3 |
| Parainfluenza virus Type 2 | | ZeptoMetrix | NATRVP-1 | 0/3 |
| Parainfluenza virus Type 3 | | ZeptoMetrix | NATRVP-1 | 0/3 |
| Respiratory Syncytial Virus A | | ZeptoMetrix | NATRVP-1 | 0/3 |
| Respiratory Syncytial Virus B | | ZeptoMetrix | NATRVP-1 | 0/3 |
| Rhinovirus 1A | | ZeptoMetrix | NATRVP-1 | 0/3 |
| Rhinovirus 1A | | ZeptoMetrix | NATPPA-BIO | 0/3 |
| RSV A2 | | ZeptoMetrix | NATPPA-BIO | 0/3 |
| S. agalactiae | Z019 | ZeptoMetrix | NATPPQ-BIO | 0/3 |
| S. aureus | MRSA;COL | ZeptoMetrix | NATPPQ-BIO | 0/3 |
| S. marcescens | Z053 | ZeptoMetrix | NATPPQ-BIO | 0/3 |
| S. pneumoniae | Z022 | ZeptoMetrix | NATPPQ-BIO | 0/3 |
| S. pyogenes | Z018 | ZeptoMetrix | NATPPQ-BIO | 0/3 |

consistency and reproducibility. The LoD determination of the Prime CovidDetect™ Rapid Detection kit was 80 copies/μL, Table 6. The amplification plots of the 24 replicate wells for SARS-CoV-2 are shown in Fig 1; specifically, Orf1, E1, N2 genes (Fig 1A), and the 18S RNA control gene (Fig 1B). The positive control (Fig 1C and 1D) was SARS-CoV-2 (heat-inactivated and spiked as previously described) at the dilution at 10,000 copies/μL (six replicates) and the No Template Control (Fig 1E and 1F) was Nuclease-Free water (6 replicates).

**Table 6.** Limit of detection (LoD) of the Prime CovidDetect™ rapid detection kit.

| Concentration | ORF1/E/N |
|---|---|
| Copies/μl | (Replicates detected) |
| 1000 | 24/24 (100%) |
| 100 | 24/24 (100%) |
| **80** | **24/24 (100%)** |
| 70 | 22/24 (91.7%) |
| 50 | 19/24 (79.1%) |
| 10 | 11/24 (41.7%) |

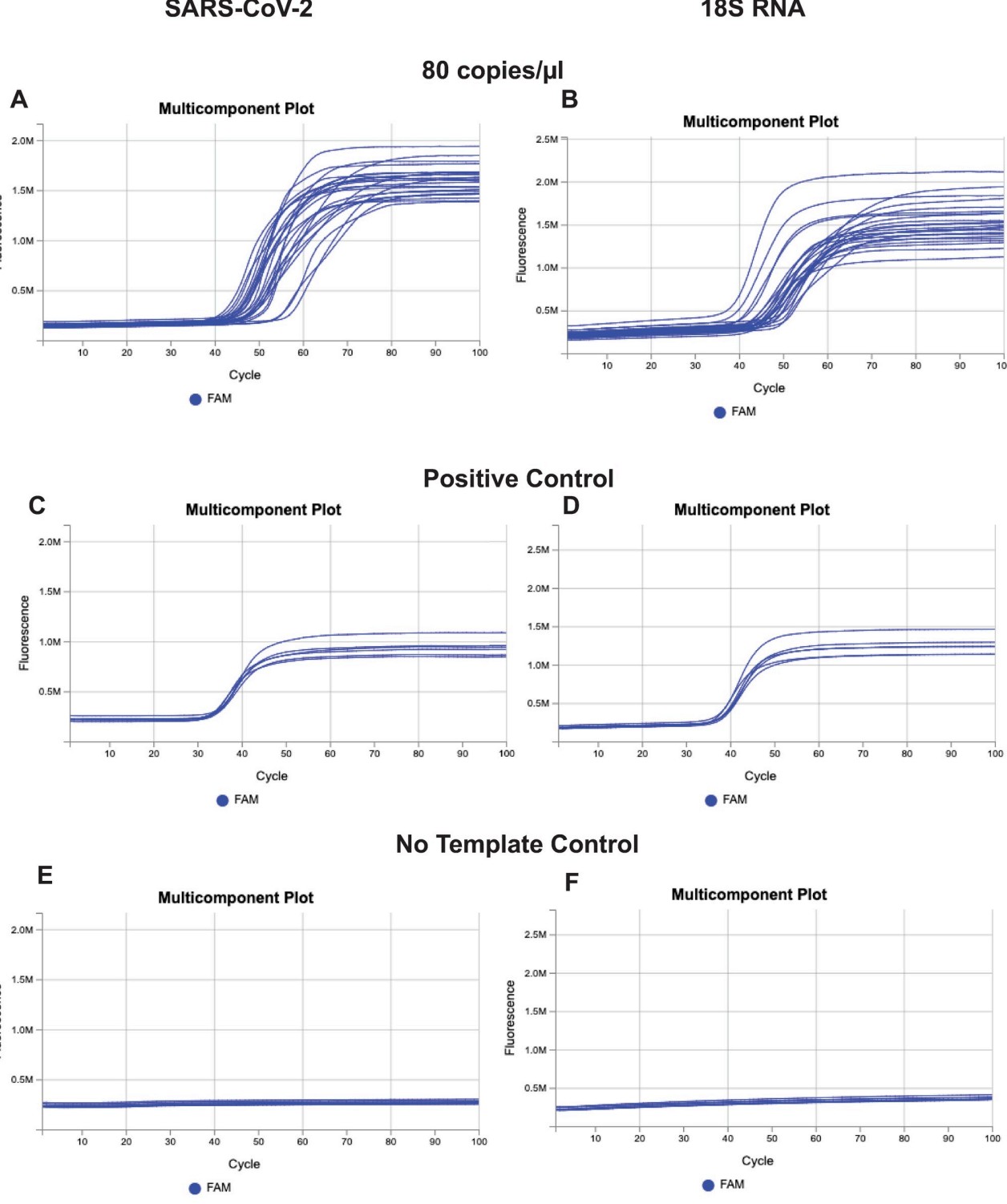

**Fig 1. Amplification plots for SARS-CoV-2 and 18S RNA.**

**Table 7. Evaluation of clinical samples and comparison to the Zymo FDA-Approved comparator assay.**

| | Prime CovidDetect™ | | FDA Approved Comparator Assay | | % Agreement |
|---|---|---|---|---|---|
| | **Positive** | **Negative** | **Positive** | **Negative** | |
| **Positive Unknown Status** | 30 | | 30 | | 100% |
| **Positive Symptomatic** | 32 | | 32 | | 100% |
| **Positive Asymptomatic** | 36 | | 36 | | 100% |
| **Negative** | | 83 | | 83 | 100% |
| **Positive Percent Agreement** | | 100% (98/98) | | | |
| **Negative Percent Agreement** | | 100% (83/83) | | | |

We then proceeded with a bioinformatic analysis to identify if the primers used within the assay were specific for SARS-CoV-2. Each primer set matched at 100% similarity against the SARS-CoV-2 Ref Seq reference genome (Wuhan-Hu-1; NC_045512.1) Table 3. In addition, an in-silico inclusivity analysis determined that the N primer set differed by one or fewer mutations for approximately 99.6% of GISAID sequences, the E primer set for 99.8%, and the ORF1 primer set for 99.8%. In total, it was determined that only 470 GISAID sequences differed by more than one nucleotide for two out of three SARS-CoV-2 primer sets, and only 41 sequences differed by more than one nucleotide for all three. Indeed, the potential for poor primer hybridization to co-occur across all three primer sets is exceedingly rare, at approximately 1 in 140,810. Our analysis revealed that the primers used to detect the genes Orf1, E, and N of the SARS-CoV-2 virus are highly specific (SARS-CoV-2 Gene Bank Reference MN908947.3) and show minimal cross-reactivity with other coronaviruses, adenoviruses, or influenza viruses Table 4. Additionally, as shown in Table 5, when the assay was used in wet testing for pathogens similar or related to SARS, no replicates were detected. Thus, both in silico and wet testing analysis showed a high specificity of the Prime CovidDetect™ Rapid Detection assay.

## Clinical evaluation

To determine the detection rate of both positive and negative confirmed nasopharyngeal swabs we assessed the performance evaluation of the Prime CovidDetect™ Rapid Detection kit on clinical samples. Samples from two sources, and from symptomatic or asymptomatic patients were used. Additionally, samples were obtained from several testing sites across the United States to consider patients' variability and potential differences in collection methods. When compared to a comparator test, approved by the FDA, the Prime CovidDetect™ Rapid Detection test showed a 100% detection rate Table 7. Positive Samples included a total of 24 out of 98 samples with a Ct value > 30 (clinically challenging samples) as quantified by the comparator assay (S2 File).

## Discussion

According to the Center for Disease and Prevention (CDC) guidelines, upper respiratory specimens, including oropharyngeal and nasopharyngeal swabs, anterior nasal and mid-turbinate nasal swabs, nasopharyngeal washes/aspirates, or nasal aspirates as well as bronchoalveolar lavage (BAL), can be used for the detection of COVID-19 in healthcare settings. Commercial SARS CoV-2 diagnostic RT-PCR kits usually detect two or more genes related to the SARS-CoV-2 virus and require the classical experimental workflow where the sample

is received in the laboratory, inventoried, RNA is extracted followed by RNA a quality control step, reverse transcription is performed, and PCR is performed. The entire process is not only labor-intensive (it can take up to more than 2 hours) but relies on expensive equipment (e.g., a quantitative PCR platform) and very often requires at least two optical filters to be able to read probes conjugated to two or more fluorophores. Additionally, the process relies on RNA extraction kits, plastic consumables, and trained laboratory scientists. Although ideal in a research laboratory setting, the entire pipeline has been revealed to be unrealistic in the context of the COVID-19 pandemic. Interestingly, a shortage of consumables and the lack of a trained workforce able to process laboratory specimens quickly and efficiently have afflicted laboratories worldwide, delaying testing. From a socioeconomic perspective, inequalities and disparities across countries have posed a challenge to COVID-19 testing. Many laboratories in challenging locations cannot afford expensive PCR equipment and highly trained staff. Rapid tests for COVID-19 based on antigen detection have been initially acclaimed as fast assays able to provide results in less than one hour, and in some cases, in less than 30 minutes. However, many concerns have been raised in the field due to collected data showing a continuous increase in false positivity rate and inaccuracies of these tests in some challenging circumstances [11,12]. Consequently, the FDA has issued an alert to healthcare providers regarding the potential for false-positive antigen results and steps to mitigate this risk (https://www.fda.gov/medical-devices/letters-health-care-providers/potential-false-positiveresults-antigen-tests-rapid-detection-sars-cov-2-letter-clinical-laboratory). Interestingly, concerns have also been raised related to false-negative results as a significant limitation of these tests. The local experience and reports to the FDA have found that antigen tests in symptomatic people are less sensitive than initially reported. In addition, these tests have much lower sensitivity when testing asymptomatic subjects. Rapid antigen tests can help quickly identify patients early in the course of SARS-CoV-2 infection when viral load is highest and who pose the greatest risk of SARS- CoV-2 transmission to others. They perform best when there is a high pre-test probability of infection (e.g., symptoms consistent with COVID-19, recent exposure to a known cause, and living/working in a setting where a high proportion of persons are infected). Thus, alternative testing methods are critically needed based on fast and reliable approaches. However, it is imperative to ensure that new candidate tests can guarantee low false positive and negative rates and ensure good specificity and sensitivity. Our study describes an RT-LAMP-based process that can quickly identify the SARS-CoV-2 RNA in clinical specimens in less than 1 hour. Because the method is based on an isothermal step, it does not require expensive PCR equipment. It could be quickly executed on a regular thermal cycler combined with a plate reader or water bath combined with a plate reader. The assay uses a fluorescent dye, and the end-point visualization can be achieved using any instrument with the following wavelength capacity: excitation 470 ±10 nm, emission 520 ±10 nm. Because the assay does not use probes but is primers based only, the manufacturing process is faster and extremely versatile as oligonucleotides can be obtained from several suppliers promptly. Perhaps, the most significant advantage of the assay described within the study is that no RNA extraction is needed. When samples are collected in the iSWAB™ (Mawi DNA Technologies) extraction-less buffer, the viral RNA is released into the collection tube and immediately available for assessment. Samples stored in the iSWAB™ do not require refrigeration and are stable at room temperature for up to twenty-one days. Both reverse transcription and LAMP reactions occur at 65 degrees Celsius, and thus, no preincubation and enzyme activation steps are required. Additionally, the assay uses a set of primers targeting three genes of the SARS-CoV-2 virus (Orf1, E1, N2) and an endogenous (18S RNA) gene. Combining three target genes (SARS-CoV-2) into the same reaction tube ensures maximum coverage and a broad detection compared to assessments

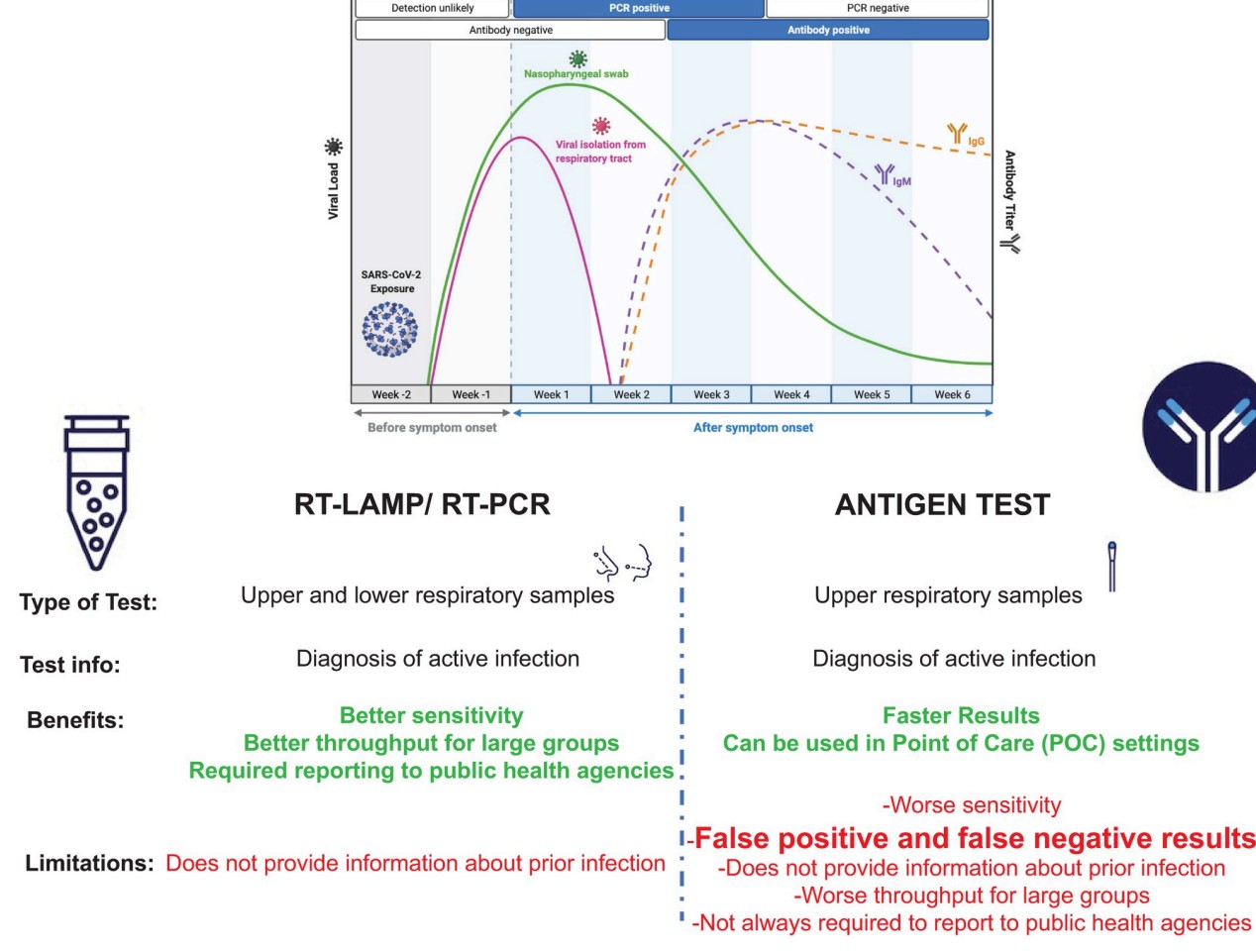

**Fig 2. Comparison of RT-LAMP/PCR versus antigen tests for the detection of SARS-CoV-2.**

based on one gene only (e.g., N1, N2). The RT-LAMP product can be monitored in a real-time fashion by the intercalating dye emission of fluorescence, or the emission signal can be detected by fluorescence readers or plate readers as an end-point assay. The specificity and sensitivity of the assay showed a high level of agreement with a standard RT-PCR FDA-approved comparator assay. Another essential advantage of the assay is that samples are analyzed in single-plex. Therefore, the probes' signal interference, the relative expression levels of targets (including endogenous controls), and the dynamic range of their expression (issues often observed in RT-PCR approaches) do not represent a concern. The Prime CovidDetect™ Rapid Detection kit based on LAMP (like PCR-based approaches) offers advantages compared to antigen tests. Negative results from a rapid antigen test are often required to be confirmed by a molecular test, and antigen tests are more likely to miss an active SARS-CoV-2 infection than molecular tests Fig 2.

Taken together, our data propose an entire pipeline (from sample collection to data visualization) that can efficiently be executed in less than 1 hour and presents a high level of versatility and adaptability not only to laboratory settings but also to impromptu testing sites

## RT-LAMP RAPID TEST

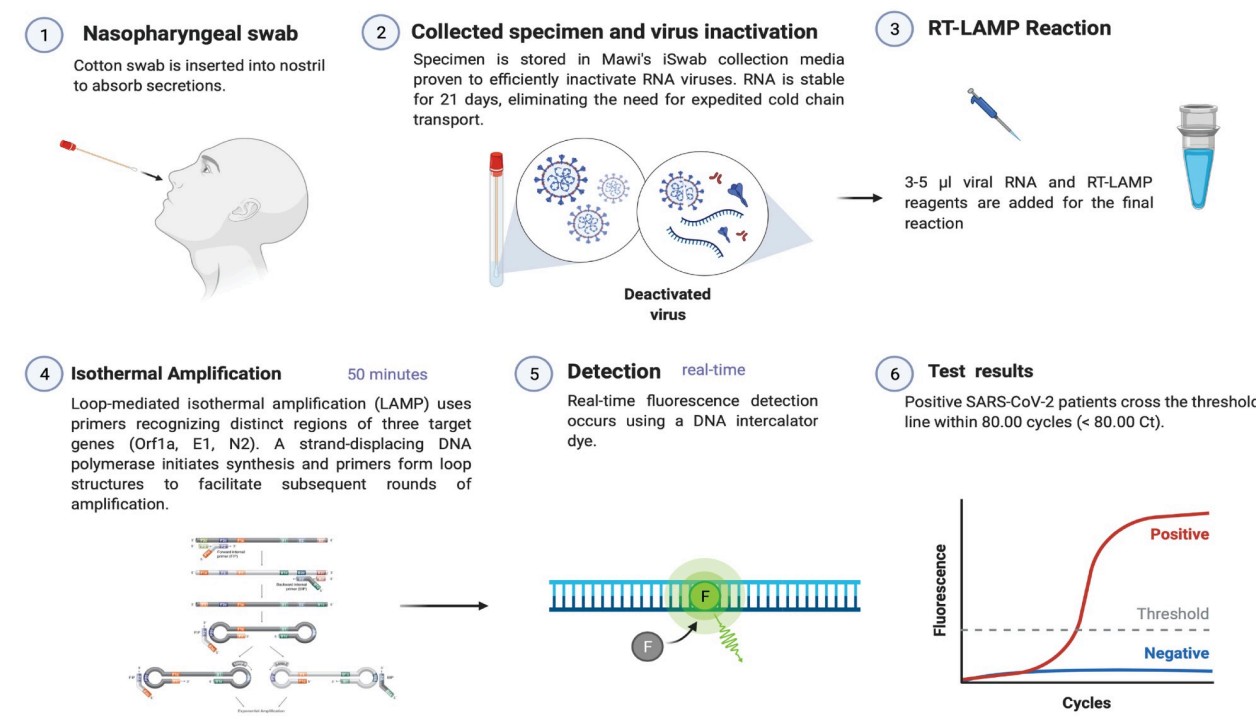

**Fig 3. RT-LAMP rapid test workflow.**

Fig 3. Compared to a standard RT-PCR pipeline, the RT-LAMP assay provides a faster turn-around for data generation, is highly versatile, scalable on-demand, requires less workforce and presents advantages compared to rapid antigen tests Fig 4. Thus, making the assay a suitable candidate for SARS-CoV-2 detection in the context of the current COVID-19 pandemic. Additionally, data collected in our laboratory has shown that the same assay can be used for the detection of other pathogens like treponema pallidum, Influenza Viruses, Hepatitis virus, and more. Although the assay represents a valuable tool for the SARS-CoV-2 detection in clinical samples, significant limitations must be considered. The detection of 18S RNA indicates that human nucleic acid is present and implies that human biological material was collected, successfully extracted, and amplified. It does not necessarily suggest that the specimen is appropriate for detecting SARS-CoV-2.—Negative results do not preclude SARS-CoV-2 infection and should not be used as the sole basis for treatment. Optimum specimen types and timing for peak viral levels during infections caused by SARS-CoV-2 are not fully determined and might impact the assay. A false-negative result may occur if a specimen is improperly collected, transported, or handled.—If the virus mutates in the LAMP target regions, SARS-CoV-2 may not be detected.—Inhibitors and other types of interference may produce false-negative results—Detection of viral RNA may not translate to causation for clinical symptoms and severity of the symptoms.—The effect of vaccines, antiviral therapeutics, antibiotics, chemotherapeutic or immunosuppressant drugs has not been evaluated.

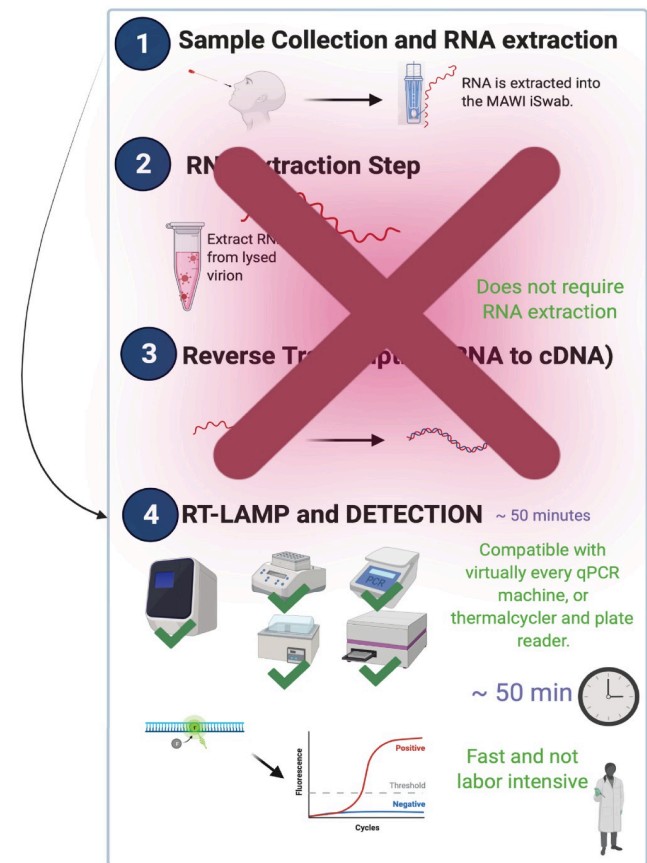

**Fig 4. RT-PCR versus extraction-less RT-LAMP workflow.**

## Supporting information

**S1 File. Programming instructions, reaction set-up, and test interpretation.**
(DOCX)

**S2 File. Ct values of positive and negative samples.**
(XLSX)

## Acknowledgments

We thank Mawi DNA technologies for providing the iSWAB™ extraction-less buffer at no cost. We also thank Hook Diagnostics Laboratories for providing clinical positive and negative samples for the clinical evaluation study. We thank prof. Kris Gunsalus (NYU Abu Dhabi) for the insightful conversations.

## Author Contributions

**Conceptualization:** Francesco Elia Marino, Arun Manoharan.

**Data curation:** Francesco Elia Marino, Arun Manoharan.

**Formal analysis:** Francesco Elia Marino, Eric Proffitt, Arun Manoharan.

**Investigation:** Francesco Elia Marino, Arun Manoharan.

**Methodology:** Francesco Elia Marino, Arun Manoharan.

**Supervision:** Francesco Elia Marino, Eugene Joseph.

**Validation:** Francesco Elia Marino.

**Writing – original draft:** Francesco Elia Marino.

**Writing – review & editing:** Francesco Elia Marino, Eugene Joseph, Arun Manoharan.

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
