## [Decision Letter · Decision Letter 0]

15 Mar 2022

PONE-D-22-04921A Rapid, Specific, Extraction-less, and Cost-Effective RT-LAMP test for the detection of SARS-CoV-2 in clinical specimens.PLOS ONE

Dear Dr. Marino,

Thank you for submitting your manuscript to PLOS ONE. After careful consideration, we feel that it has merit but does not fully meet PLOS ONE’s publication criteria as it currently stands. Therefore, we invite you to submit a revised version of the manuscript that addresses the points raised during the review process.

We look forward to receiving your revised manuscript.

Kind regards,

Ruslan Kalendar

Academic Editor

PLOS ONE

Journal Requirements:

"The author(s) received no specific funding for this work. The work has been funded with private funding at Prime Discoveries. "

"Francesco Elia Marino, Eugene Joseph, and Arun Manoharan are full time employee at Prime Discoveries. "

We note that one or more of the authors are employed by a commercial company: Prime Discoveries

5. Please include a copy of Table 6 which you refer to in your text on page 16.

Reviewers' comments:

Reviewer's Responses to Questions

**Comments to the Author**

1. Is the manuscript technically sound, and do the data support the conclusions?

Reviewer #1: Yes

Reviewer #2: Partly

Reviewer #3: Yes

2. Has the statistical analysis been performed appropriately and rigorously? 

Reviewer #1: N/A

Reviewer #2: N/A

Reviewer #3: N/A

3. Have the authors made all data underlying the findings in their manuscript fully available?

Reviewer #1: Yes

Reviewer #2: Yes

Reviewer #3: Yes

4. Is the manuscript presented in an intelligible fashion and written in standard English?

Reviewer #1: Yes

Reviewer #2: Yes

Reviewer #3: Yes

5. Review Comments to the Author

Reviewer #1: 

The manuscript reports developed a RT-LAMP test for SARS-CoV-2. They used iSWAB extraction-less buffer to skip the extraction step and perform RT-LAMP test. Three targets ORF1 a/b, E, and N genes are used and 18S as internal control.

For clinical evaluations, 30 positive and 34 negatives by real time rt-PCR were used. The ct values of these 30 positives are missed in the manuscript and is a required to evaluate how sensitive of RT-LAMP. A table is needed to present the Ct of those positive samples and with RT-LAMP result. If these pos samples had lower ct values, weak positive samples should be included for evaluations.

Tables 3 to 5 can be moved to supplemental parts.

Reviewer #2: 

The manuscripts evaluates an RT LAMP approach for SARS-CoV-2 detection. The technology is getting a lot of attention and these studies are welcome. overall the study is straightforward. However, the conclusions are based on a small number of clinical samples not well characterized and stratified. unless this is provided the study is deceptive by showing 100% sensitivity and 100% ppv/npv with a calculated LoD of 80 cp/uL, which is above the theshold of most RT PCR kits. a minimal requirement would be to include genomes/uL or Ct values of the tested samples. a much better study would require a portion of 25% of positive samples Ct>30 (low viral load).

below some specific comments:

define cost-effective, the technology uses as read-out the real-time thermocycler so the cost is quite high - is it necessary? could it be colorimetric?

line 47 = causing the COVID-19 pandemic.

line 75 … identify people with COVID-19 who do not have symptoms and slow the spread of SARS-CoV-2 = Identify asymptomatic individuals carrying the virus.

line 78-88 quite unspecific, need to focus on published papers showing advantage of RT LAMP, use of colorimetric readout, use of a heat-block instead of thermocycler et etc (i.e. https://doi.org/10.1016/j.eclinm.2021.101101)

line 128 temperature not indicated, 65 °C??

line 173, threshold is arbitrary, why 80 cycles? clarify

line 177 update on current variants

line 244 and 253 ethics of human samples, some purchased some collected, clarify

LoD at 80 cp/ul is higher than RT PCR so I expect low viral loads not to be detected

stats not indicated, not necessary for these values at 100% but in case of adding more specimens then a contingency table with stats would be required.

Reviewer #3: 

The submission by FE Marino et al is clear and easily readable. It is worthy of publication subject to some modifications or clarifications:

line 93: temperatures and reaction times are key elements in molecular biology techniques. However, not all laboratories are equally accurate in obtaining temperatures. Have the authors tested a temperature variation around 65 degrees and a time variation around 50 minutes?

line 96: It is interesting that the authors used iSWAB Extraction-less buffer swabs. However, during the pandemic we experienced many stock-outs (the authors mention it lines 385 to 387) and swabs were not spared. It would be interesting if the authors could test other types of swabs or even home-made swabs.

line 271: no need to repeat the LoD definition

line 289: table 6 not table 5

line 372: remove the S from DISCUSSIONS

line 409: the authors claim a result in one hour. This should be balanced as there is a long pre-analytical phase of sample deposition on the QS5 plate which should not be forgotten

line 456: is the detection of 18S ribosomal RNA not sufficient to validate the absence of inhibitors?

Table 4: It would be interesting if the viruses were mentioned next to the GenBank identification

Table 5: group pathogens by taxonomy; what is the enterovirus proposed by ZpetoMetrix, ditto for influenza B virus; have you tested a parainfluenza 4, what is the meaning of COL

6. PLOS authors have the option to publish the peer review history of their article (what does this mean?). If published, this will include your full peer review and any attached files.

Reviewer #1: No

Reviewer #2: No

Reviewer #3: **Yes: **Dr Jean-Michel MANSUY (MD)

---

## [Author Response · Author response to Decision Letter 0]

19 Mar 2022

Rebuttal Letter to Editor and Reviewers.

Response to Editor

We note that you have included the phrase “data not shown” in your manuscript. Unfortunately, this does not meet our data sharing requirements. PLOS does not permit references to inaccessible data. We require that authors provide all relevant data within the paper, Supporting Information files, or in an acceptable, public repository. Please add a citation to support this phrase or upload the data that corresponds with these findings to a stable repository (such as Figshare or Dryad) and provide and URLs, DOIs, or accession numbers that may be used to access these data. Or, if the data are not a core part of the research being presented in your study, we ask that you remove the phrase that refers to these data.

-We have removed “data not shown” in the updated version of the manuscript and only referred to data available provided within the manuscript.

Please include a copy of Table 6 which you refer to in your text on page 16

-As correctly stated by Reviewer #3, this was a typo. All the tables were in the manuscript, but one portion of the text referred to the wrong table. We have amended the text and corrected the mistake. 

Response to Reviewer 1

The manuscript reports developed a RT-LAMP test for SARS-CoV-2. They used iSWAB extraction-less buffer to skip the extraction step and perform RT-LAMP test. Three targets ORF1 a/b, E, and N genes are used and 18S as internal control.

For clinical evaluations, 30 positive and 34 negatives by real time rt-PCR were used. The ct values of these 30 positives are missed in the manuscript and is a required to evaluate how sensitive of RT-LAMP. A table is needed to present the Ct of those positive samples and with RT-LAMP result. If these pos samples had lower ct values, weak positive samples should be included for evaluations.

Tables 3 to 5 can be moved to supplemental parts.

--Dear reviewer, we had a total of 98 positive samples (combined) and 83 negative samples (combined) included within this study. All positive and all negative samples were analyzed by RT-PCR and Ct values are available for all the samples. Please see our updated supplemental material where we have reported the Ct values for all the positive and negative samples. 

Clinically challenging samples with a Ct value > 30 represented 24.49% of positive samples (24 out of 98 positive samples). 

Thanks for your time and your comments. 

Response to Reviewer 2

The manuscript evaluates an RT-LAMP approach for SARS-CoV-2 detection. The technology is getting a lot of attention and these studies are welcome. overall the study is straightforward. However, the conclusions are based on a small number of clinical samples not well characterized and stratified. unless this is provided the study is deceptive by showing 100% sensitivity and 100% ppv/npv with a calculated LoD of 80 cp/uL, which is above the threshold of most RT PCR kits. a minimal requirement would be to include genomes/uL or Ct values of the tested samples. a much better study would require a portion of 25% of positive samples Ct>30 (low viral load).

--Dear reviewer, we have provided the Ct values in the supplemental material of the resubmitted manuscript. Our cohort of samples included 24.49% of positive samples with Ct values > 30 (24 positive samples out of 98 positive samples). 

--Also, we want to specify some important points 

--Our combined cohort of the positive samples was 96 positive samples and 83 negative samples. 

--When we started our study, we have observed an important and perhaps, expected effect: samples decay and a significant drop in viral load over time. Many groups around the world have performed comparisons using Ct values provided by the vendor or testing laboratories (at the moment of testing) and then compared the performance to their proposed assay (very often performed many days later if not weeks later). This is not a fair comparison because Ct values collected days before comparison, for example, are not really informative (samples will inevitably degrade over time and the drop in sensitivity at that point is unclear if due to the assay itself or other external factors like viral decay, storage conditions, etc.).For our comparison, we decided to re-test every positive and negative sample in our laboratory the same day (or the day before) when the LAMP assay was executed using the FDA-approved comparator assay (Zymo). Doing so we had a fair comparison between technologies and viral decay did not affect the performance. This could explain our very high sensitivity rate. If a sample failed to be detected the day of testing in our laboratory using the RT-PCR FDA approved test (independently from what was previously reported by the vendor or testing lab) such sample was not used for the comparison because “not viable” at the moment of testing, in other words not viable at the moment of plate loading. Additionally, we use three targets for SARS-CoV-2 in our LAMP Assay (Orf1, E1, N2) vs. many other assays focused on N1/N2 genes only or N1 only. We will increase our pool of samples in the future and determine how sensitivity changes with a very large group of samples. However, we believe that our assay is robust in terms of sensitivity according to the data we have collected. 

--We also would like to bring to your attention that we have asked for an external evaluation assessment on a total of 141 samples (55 positive samples and 83 negative samples) collected by healthcare providers from patients seeking SARS-CoV-2 testing and previously tested at New York University in Abu Dhabi using an implementation of the CDC 2019-nCoV Real-Time PCR Test. The NYU Abu Dhabi samples were tested in an automated high throughput setup using the Chemagic 360 automated RNA extraction and pipetting was tested with Agilent Bravo automated liquid handlers. All results generated in NYU Abu Dhabi using our LAMP assay were concordant with the RT-qPCR results obtained in the same testing lab in NYU Abu Dhabi. Our decision to remove this additional cohort of samples (external validation in Abu Dhabi) from the current manuscript was because I did not perform quantitative PCR and comparison in our laboratory and, those samples were not analyzed with the same comparator assay (Zymo Research) I used in our laboratory. Nevertheless, the NYU Abu Dhabi laboratory reported the following data to us:

Nasopharyngeal Swabs Comparator - EUA Authorized Assays

 Positive Negative Total

Prime COVID-19 High Throughput Assay Result Positive 53 0 53

 Negative 0 88 88

 Total 53 88 141

Positive Percent Agreement 100% (53/53); 97.14% - 100.00%*

Negative Percent Agreement 100.0% (88/88); 97.74% - 100.00%*

* 95% confidence interval. Jovanovic B. D., & Levy, P. S. (1997). A Look at the Rule of Three.

---Thus, we think that this proposed assay and pipeline performs particularly well in clinical specimens. 

below some specific comments:

define cost-effective, the technology uses as read-out the real-time thermocycler so the cost is quite high - is it necessary? could it be colorimetric?

-- Cost-effective is specifically referred to as the cost per reaction for the RT-LAMP (oligonucleotides, dye, and enzyme only) compared to a quantitative PCR approach based on oligonucleotides, probes, and enzyme. The dye’s cost itself is 1/3 lower than fluorescent probes. 

-- Additionally, cost-effectiveness refers to the elimination of RNA extraction kits (viral RNA extraction kits are much more expensive than the iSWAB extraction-less buffer).

-- As per our discussion, our assay does not necessarily require a RT-thermocycler. We have provided instructions within the supplemental materials to perform the assay with a PCR cycler (or heat block) combined with a plate reader (the only requirement is the ability of such plate reader to detect the FAM/SYBR green fluorescence – virtually every plate reader on the market should have this reading capability). 

-- We have initially considered and tested in our laboratory the colorimetric approach but after extensive research and development assessments, we have concluded that the colorimetric approach is significantly limited in terms of ambiguity when it comes to data interpretation. Our initial validation data were discussed with New England Biolabs (pioneer company in LAMP development) and we came to the conclusion that the colorimetric approach requires extra steps of control and is easily affected by changes in pH. For example, we have tested the colorimetric approach on saliva, and we have observed significant effects on pH, therefore, affecting the colorimetric performance and reliability. We, therefore, focused our resources on the RT-LAMP approach combined with a more objective and measurable optical readout. However, we understand that in challenging locations around the world the colorimetric approach is still a valid option. 

 line 47 = causing the COVID-19 pandemic.

--we have amended the text. 

line 75 … identify people with COVID-19 who do not have symptoms and slow the spread of SARS-CoV-2 = Identify asymptomatic individuals carrying the virus.

--we have amended the text. 

line 78-88 quite unspecific, need to focus on published papers showing advantage of RT LAMP, use of colorimetric readout, use of a heat-block instead of thermocycler et etc (i.e. https://doi.org/10.1016/j.eclinm.2021.101101)

-- We have amended the text and discussed why we did focus on a non-colorimetric approach. 

-- Additionally, we would like to draw your attention to the fact that our pipeline, and its proposed novelty, is not only based on the RT-LAMP approach itself (which is not novel) but on the possibility to bypass the RNA extraction step (a significant limiting factor in testing for laboratories and impromptu testing sites). E.g., at the beginning of the pandemic obtaining a Qiagen Viral RNA extraction kit was extremely difficult. Bypassing the RNA extraction step, without significantly compromising the limit of detection has represented the most difficult technical challenge in the field. We believe that the FDA-approved MAWI extraction-less buffer when combined with an RT-LAMP approach like the one described in our study is an extremely valuable tool for testing and screening and it is much better than classic and laborious methods like heat inactivation or samples pre-treatment with proteinase K. Additionally, we draw your attention to the implication of storage and samples’ refrigeration requirements. For example, the study you cited by Baba et al. 2021, is primarily based on samples collected in VTM (viral transport media). Extraction-less protocols in VTM or UTM are virtually impossible to achieve without compromising performance. In that case, samples need to be analyzed within 48 hours and refrigeration is required. One of the main advantages of the iSWAB extraction-less buffer is that this buffer stabilizes the sample at room temperature for up to twenty-one days and therefore, not only the RNA extraction step can be skipped but also the need for refrigeration can be bypassed (MAWI DNA technologies). https://mawidna.com/the-iswab-advantage/. During research and development, we have tested several viral transport media but the iSWAB extraction-less buffer is without a doubt the best companion for our RT-LAMP assay when it comes to performance and LoD. 

line 128 temperature not indicated, 65 °C??

--We indicated the temperature settings at line 93 but not on line 128. We have amended the text to reflect the temperature settings with the revised version of the manuscript. 

line 173, threshold is arbitrary, why 80 cycles? Clarify

--The threshold was established based on false-positive amplification (a known RT-LAMP artifact described since its original development) observed for both SARS-CoV-2 and 18S RNA detection. Such artifacts occur at amplification cycles higher than 80 (for our specific protocol where each cycle is of 30 seconds). We have added this info within the text of the revised manuscript. 

line 177, update on current variants

--We have updated the bioinformatic analysis as you suggested according to the deposited data (March 16, 2022) as stated within the manuscript. 

line 244 and 253 ethics of human samples, some purchased some collected, clarify 

--We have initially purchased the first set of samples (30 positives and 34 negatives) from the vendor LEE BioSolutions (this was at the very beginning of the pandemic when accessing clinical specimens was extremely difficult – the year 2020). However, no symptomatic status was provided by the vendor because the information was simply not collected at that time. To expand the cohort of samples and include symptomatic status, we later added a second set of samples obtained from a diagnostic lab with available symptomatic status (the year 2022).

LoD at 80 cp/ul is higher than RT PCR so I expect low viral loads not to be detected stats not indicated, not necessary for these values at 100% but in case of adding more specimens then a contingency table with stats would be required.

-- As shown within our data the assay was able to detect samples with a Ct value higher than 30. We would expect issues with the detection of samples with very low viral load (e.g. Ct values >36-37. To assess samples with a very low viral load we would have needed to include samples with Ct values higher than 35-37. However, most RT-PCR kits have a cut-off at 35-37 Ct, and samples with borderline Ct values are usually considered inconclusive or recommended for re-testing (therefore not usually reported). Originally, we reached out to vendors like LEE BioSolutions to seek such highly challenging samples (Ct >36) but samples with such Ct values were not available for the above-mentioned reasons. Additionally, we agree with the reviewer that the RT-LAMP assay itself cannot have the same sensitivity as a quantitative RT-PCR assay due to the different nature and chemistry between the two methodologies. However, as we have shown in our study, together with data reported from other groups, RT-LAMP combined with a reliable method of RNA extraction (like our iSWAB extraction-less buffer) can have significant advantages in the field for testing clinical samples which supposedly have a viral load much higher than 80 copies/ul (e.g., active infection, transmissibility potential, etc.)

-- We did not increase the number of positive or negative samples compared to our original submission and therefore, we did not need to perform statical analysis as you have already suggested. 

--Thank you for your time and insights. 

Reviewer #3:

The submission by FE Marino et al is clear and easily readable. It is worthy of publication subject to some modifications or clarifications:

line 93: temperatures and reaction times are key elements in molecular biology techniques. However, not all laboratories are equally accurate in obtaining temperatures. Have the authors tested a temperature variation around 65 degrees and a time variation around 50 minutes?

--We have tested our LoD data on different types of equipment (see below). We have not observed effects on the results. However, our validation studies for the current manuscript were all conducted on the QuantStudio version 5 for consistency and reproducibility (as stated in our Material and Methods section). The in-silico primers design predicts primers’ performance in the range of 60-65 degrees Celsius and therefore, small fluctuations around 65 degrees should not impact the results. However, our recommended protocol remains at 65 degrees. 

- QuantStudio version 3

- QuantStudio version 5

- QuantStudio version 6

- QuantStudio version 7 pro

- QuantStudio version 6 pro

- Veriti Thermal Cyclers and Plate Reader (Tecan 200 Pro) 

line 96: It is interesting that the authors used iSWAB Extraction-less buffer swabs. However, during the pandemic we experienced many stock-outs (the authors mention it lines 385 to 387) and swabs were not spared. It would be interesting if the authors could test other types of swabs or even home-made swabs.

--We agree and this is a very valid point. This is the exact reason why MAWI DNA technologies started the production of their own swabs engineered to capture more biological material and to be able to support the demand of swabs + iSWAB Extraction-less buffer. Additionally, the MAWI DNA swab is engineered differently compared to classic nasal swabs (I am providing a picture for your reference); the swab is a 100% plastic injection molded swab and outperforms flocked swabs in many applications. Due to the 100% plastic molded construction, supply chain issues are close to nonexistent according to the manufacturer (https://mawidna.com/coming-soon-nextswab-next-generation-swab-from-mawi-dna-technologies/). To the best of our knowledge, MAWI DNA technologies does not forecast a shortage of their swabs at the current moment or experienced shortages at any time since the beginning of the current pandemic. (https://mawidna.com/uncategorized/mawi-maintains-robust-stock-of-covid-testing-supplies-with-usa-based-uninterrupted-supply-chain/

line 271: no need to repeat the LoD definition 

--We amended the text. 

line 289: table 6 not table 5

--Thank you, that was a typo. We corrected it. 

line 372: remove the S from DISCUSSIONS

--We amended the text. 

line 409: the authors claim a result in one hour. This should be balanced as there is a long pre-analytical phase of sample deposition on the QS5 plate which should not be forgotten

--For comparison purposes, we have excluded the loading component-time as this step would equally increase the RT-LAMP time and the PCR time. Thus, it would not affect the overall advantage in terms of time-saving. Additionally, the plate loading step is highly variable and can be fully automatable. Therefore, it could take from 5 to 20 minutes based on the laboratory budget, staff, and equipment (e.g., automated pipettors, pre-loaded reagents in analytical plates, robotic automation, etc.). Once the experimental pipeline is defined the plate mapping and instrument initialization could take as low as 5 minutes if everything is prepared beforehand (of course this would require a certain level of organization). 

line 456: is the detection of 18S ribosomal RNA not sufficient to validate the absence of inhibitors? 

--Theoretically, yes. However, we prefer to be cautious and state this potential limitation in the discussion. 

Table 4: It would be interesting if the viruses were mentioned next to the GenBank identification

--Thank you for the suggestion. We have included designations. 

Table 5: group pathogens by taxonomy; what is the enterovirus proposed by ZpetoMetrix, ditto for influenza B virus; have you tested a parainfluenza 4, what is the meaning of COL

--The specific strains for the Enterovirus and the Influenza B Virus are not specified by the vendor for this panel https://www.zeptometrix.com/media/documents/PINATRVP-1.pdf

--We did not test parainfluenza 4 (wet testing) because not originally provided with the Zeptometrix Panel. However, the in-Silico data is shown as NC_021928.1 (Human parainfluenza virus 4a viral cRNA, complete genome, strain: M-25). 

--We have amended the Strain nomenclature for COL to reflect the full nomenclature MRSA; COL (as provided by the vendor). The strain refers to S.aureus. 

--Thank you for your time and suggestions. These were greatly appreciated.

---

## [Decision Letter · Decision Letter 1]

25 Mar 2022

A rapid, specific, extraction-less, and cost-effective RT-LAMP test for the detection of SARS-CoV-2 in clinical specimens.

PONE-D-22-04921R1

Dear Dr. Marino,

We’re pleased to inform you that your manuscript has been judged scientifically suitable for publication and will be formally accepted for publication once it meets all outstanding technical requirements.

Kind regards,

Ruslan Kalendar

Academic Editor

PLOS ONE

Reviewers' comments:

Reviewer #2: the authors provided a sound revision of their manuscript addressing all the points raised

Reviewer #3: The paper now sounds well. The description of a simple, unexpensive, rapid, sensitive and specific molecular assay for the virological diagnosis of COVID-19 is of importance especially for LMIC.

I thank the authors for taking into account my suggestions.

Dr Jean-Michel MANSUY (MD)

---

## [Editor Report · Acceptance letter]

1 Apr 2022

PONE-D-22-04921R1 

A rapid, specific, extraction-less, and cost-effective RT-LAMP test for the detection of SARS-CoV-2 in clinical specimens. 

Dear Dr. Marino:

I'm pleased to inform you that your manuscript has been deemed suitable for publication in PLOS ONE. Congratulations! Your manuscript is now with our production department. 

Kind regards, 

on behalf of

Professor Ruslan Kalendar 

Academic Editor

PLOS ONE